# Region-Specific Remote-Sensing Models for Predicting Burn Severity, Basal Area Change, and Canopy Cover Change following Fire in the Southwestern United States

Alicia L. Reiner [1,*](ID), Craig Baker [1], Maximillian Wahlberg [2], Benjamin M. Rau [3] and Joseph D. Birch [4](ID)

1    Geospatial Technology and Applications Center, USDA Forest Service, Salt Lake City, UT 84138, USA
2    Pacific Northwest Region, USDA Forest Service, Portland, OR 97204, USA
3    Timber and Watershed Research Laboratory, Northern Research Station, USDA Forest Service, Parsons, WV 26287, USA
4    Department of Plant, Soil, and Microbial Sciences, Michigan State University, East Lansing, MI 48824, USA
*    Correspondence: alicia.reiner@usda.gov; Tel.: +1-530-559-4860

**Abstract:** Estimates of burn severity and forest change following wildfire are used to determine changes in forest cover, fuels, carbon stocks, soils, wildlife habitat, and to evaluate fuel and fire management strategies and effectiveness. However, current remote-sensing models for assessing burn severity and forest change in the U.S. are generally based on data collected from California, USA, forests and may not be suitable in other forested ecoregions. To address this problem, we collected field data from 21 wildfires in the American Southwest and developed region-specific models for assessing post-wildfire burn severity and forest change from remotely sensed imagery. We created indices (delta normalized burn ratio (dNBR), relative delta normalized burn ratio (RdNBR), and the relative burn ratio (RBR)) from Landsat and Sentinel-2 satellite imagery using pre- and post-fire image pairs. Burn severity models built from southwest U.S. data had clear advantages compared to the current California-based models. Canopy cover and basal area change models built from southwest U.S. data performed better as continuous predictors but not as categorical predictors.

**Keywords:** burn severity; southwestern U.S.; remote sensing; composite burn index (CBI); RdNBR; dNBR; RBR

## 1. Introduction

Fire is a key ecosystem process in the southwest U.S., and quantifying its severity provides the foundation for predicting and managing a variety of social and ecological processes. From 2012 to 2019, approximately 100,000 to 200,000 ha burned annually in Arizona (AZ) and New Mexico (NM) combined; however, 900,000 ha burned in 2011 and 40,000 ha in 2020 [1]. Large fire years in the Southwest often correlate with drought and the La Nina phase of the Southern Oscillation [2,3]. The Southwest historically had an abundance of low severity, frequent fires that maintained open forests with a healthy understory grass component. Fire frequency typically decreases in the Southwest with elevation and moisture, and fires often become stand-replacing at higher elevations, occurring mainly during extreme drought [4]. Wildfire events with uncharacteristically high intensity or extent can result in the degradation of ecosystem services [5,6]. The quantification of burn severity can prime the understanding of changes to fuels, soils, and wildlife habitat [7]. Characterizing burn severity and forest change at landscape scale also informs post-fire decision making and improves understanding of the carbon, financial, and ecological impacts of fire [8–11]. Phenological, fire regime, and forest structure differences in the Southwest present challenges that could confound modeling burn severity compared to other western U.S. forests.

The USDA Forest Service (USFS) Geospatial Technology and Applications Center provides burn severity (composite burn index (CBI)) and forest change estimates (percent

basal area loss (BA) and percent canopy cover loss (CC)) for large fires on forested lands in the U.S. through the Rapid Assessment of Vegetation Condition after Wildfire (RAVG) program. The program provides these products in two timeframes: an "Initial Assessment" (IA) based on imagery acquired within a few weeks after fire containment and an "Extended Assessment" (EA) using post-fire imagery from approximately one year post fire (near the following peak of greenness) [12]. The IA assessment timeframe allows for first-order fire effects to be determined more clearly; however, the EA assessment timeframe favors estimation of survivorship and delayed mortality [12]. The IA RAVG burn severity products are often favored by land and fire managers looking for burn severity data during the current fire season; however, most burn severity field campaigns find IA field data collection to be logistically difficult to accomplish, as the timeframe between fire extinction and monsoon or snowfall is often narrow, and hence build models with EA data.

The standard RAVG estimates are calculated from models relating field-based measures of burn severity to the relative delta normalized burn ratio (RdNBR) [13]. The RdNBR is based on the normalized burn ratio (NBR), which is the difference of the near-infrared (NIR, e.g., Landsat 8 OLI band 5, 0.851–0.879 µm) and the short-wave infrared (SWIR2, e.g., Landsat 8 OLI band 7, 2.107–2.294 µm) spectral bands, divided by the sum of the two. The NIR is lower when less green vegetation is present, and the SWIR increases when more ash and char are present [14]. The delta normalized burn ratio (dNBR) is the difference of the pre-fire NBR and post-fire NBR, multiplied by 1000 [12]. The RdNBR relativizes the dNBR using the pre-fire NBR to moderate the effects of low vegetation pre-fire [13]. Parks et al. (2014) [15] made a further adjustment to RdNBR to assure the denominator is always greater than zero, thus developing the relative burn ratio (RBR). Additionally, the dNBR, and its derivatives (RdNBR and RBR), can utilize an offset in the calculations, which accounts for possible phenological differences between pre- and post-fire dates [16]. The offset is the average dNBR within one or more relatively homogeneous unburned areas outside each fire.

Current RAVG models were developed from data gathered in the Sierra Nevada, northern California, and southern Oregon, USA [17], yet are routinely applied across the conterminous U.S. Concern has arisen that the accuracy of these estimates may vary geographically given ecological differences in both pre- and post-fire conditions across regions [18]. Models derived from a single region and forest type could fail to adequately represent phenological, fire regime, and forest structure differences found in other regions, including the neighboring Southwest, USA. As in the Sierra Nevada, frequent, low-severity fire regimes can prevail in the dry and mixed conifer forest types of the Southwest [19]. Like California, high-severity fires in the Southwest continue to increase, especially in high-elevation areas [20,21]. Unlike the Mediterranean climate of the Sierra Nevada, however, precipitation in the Southwest occurs bimodally, with the precipitation occurring during the summer monsoon around July and August and with synoptic events in winter [4,22]. Wildfire is generally more widespread early in the summer with a typical fire season peaking just before the onset of heavy precipitation associated with the summer monsoon [4]. This bimodal precipitation regime can affect burn severity modeling if monsoonal precipitation results in ash loss or rapid green-up of grasses, sprouting shrubs, or trees, which can moderate the changes in NBR as compared to areas with much less summer moisture. Additionally, the sparse canopy cover of Southwest woodlands and the abundance of grasses make modeling canopy cover and basal area changes in these vegetation types challenging with satellite imagery, as the understory signature may overwhelm and mute the overstory signature. The RdNBR index is prone to producing extreme values when pre-fire vegetation is extremely low, which can appear as outliers but do not necessarily describe drastic change due to fire [15].

The purpose of this study was to determine if models created specifically for the Southwest would produce better burn severity and forest change estimates than the current models by comparing the predictive accuracy of both sets of models [23]. An analogous project developed models specific to the Pacific Northwest region [24]. Region-specific

models may be able to better address localized ecological dynamics by better fitting data distributions of response variables to predictor variable ranges for the region. Canopy cover and basal area change models relying on satellite indices such as RdNBR have the potential for lower accuracy in areas of lower tree canopy cover, such as the Southwest [17], because the understory can dilute the spectral signature of the trees. For this project, we evaluated models including two additional burn severity indices, dNBR and RBR, topographic and ecological variables, and various model forms, in addition to the use of region-specific field and photographic interpretation training data in efforts to improve the predictive capacity of models.

## 2. Methods

### 2.1. Site Locations

Vegetation in the Southwest varies from low-elevation shrub steppe to chapparal, woodland, and montane conifer forests at higher elevations. Woodlands often consist of Pinyon pine (*Pinus edulis* Engelmann) and *Juniperus* species, while forests range from dry forests dominated by Ponderosa pine (*Pinus ponderosa* var. *arizonica* EngelmannShaw), at times including Gambel oak *(Quercus gambelii* Nuttall), to mixed conifer forests including white fir (*Abies concolor* Gordon and Glendinning-Hildebrand) and interior Douglas fir (*Pseudotsuga menziesii* var. glauca Mayr-Franco) and at the highest elevations mesic species including Engelmann spruce *(Picea engelmannii* Engelmann) and Rocky Mountain subalpine fir *(Abies bifolia* A. Murray bis.). The lower elevation forests and woodlands typically have more open tree canopies and a mix of grass, herbaceous, tree litter, and dead woody material making up the surface fuels, whereas the upper elevations typically have dense conifer fuels with canopies that extend to the forest floor to meet surface fuels made up largely of conifer litter. Microclimate, based on topographic position, elevation, and aspect, plays a major role in the distribution of vegetation in the Southwest, with topo-edaphic climax communities shifting based on aspect and energy setting even within the same elevation and precipitation bands [22]. The primary target vegetation types for this study include the montane conifer forested types most subjected to forest management practices. These include multiple ponderosa pine types, mixed mesic and wet mixed conifer types, and spruce–fir types. Taxonomic nomenclature follows the *Flora of North America* (eds. 1993+) [25].

### 2.2. Field Sampling

A Southwest-specific field dataset was obtained to train models to satellite imagery. Field sampling design and plot placement followed Key and Benson (2006) [12] and Miller et al. (2009) [17]. Fires in AZ and NM from 2017 and 2018 were chosen for field sampling (Figure 1, Table 1). We considered candidate fires for RAVG product development if they included large portions of federal land and were accessible via roads. Sampling was carried out at even intervals along roads or trails at a target density of 15–30 plots per fire. Circular plots measured 30 m in diameter and were located at least 500 m apart, ≥100 m from roads or trails, in areas with >10% tree cover, and in areas of homogeneous burn severity, preferably 60 m × 60 m [12]. We collected location data at the center of each plot using both a Garmin Glo and a Trimble GeoXH GPS and averaged location data to improve accuracy and reliability [26]. We included unburned plots ($n$ = 67) as 20% of the entire dataset to ensure that models span the full range of wildfire severities [27]. We removed several plots from our analysis that had received post-wildfire management (e.g., salvage logging) between the time of the fire and our 1-year post-wildfire imagery. Our final plot sample size was 337.

To assess the composite burn index (CBI), we used a CBI questionnaire to generate a composite score for each plot, following Key and Benson (2006) [12]. The composite score accounts for fire effects on each of five strata: substrate, low understory, taller understory, midstory trees, and big trees, based on ocular estimates of scorch, consumption, and other changes related to fire [12].

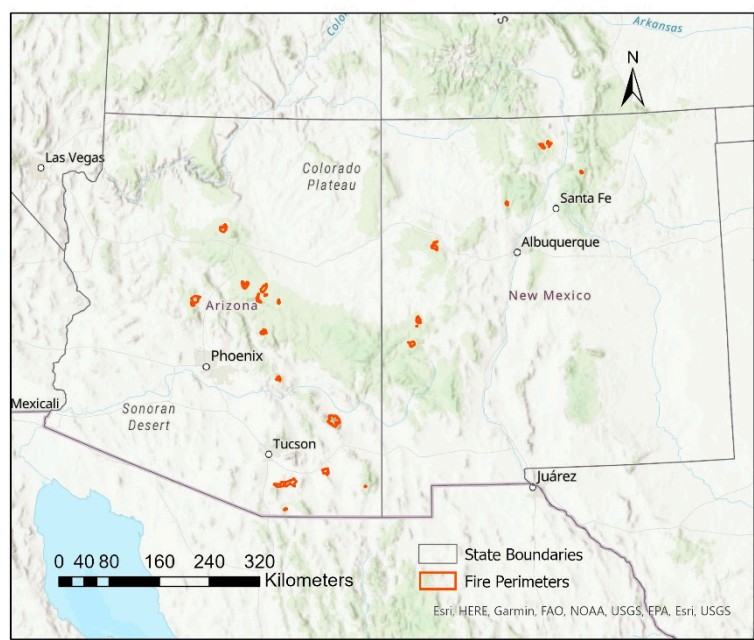

**Figure 1.** Locations of 2017 and 2018 fires where field plots were located.

**Table 1.** Locations and number of plots on each fire sampled.

| Fire Name | National Forest | State | Ignition Date | Year Sampled | Plots |
|---|---|---|---|---|---|
| Bear | Tonto | AZ | 16 Jun 2018 | 2019 | 17 |
| Blue Water | Cibola | NM | 12 April 2018 | 2019 | 22 |
| Diener Canyon | Cibola | NM | 12 April 2018 | 2019 | 25 |
| Sardinas Canyon | Carson | NM | 24 June 2018 | 2019 | 20 |
| Tinder | Coconino | AZ | 27 April 2018 | 2019 | 25 |
| Venado | Santa Fe | NM | 20 July 2018 | 2019 | 19 |
| 33 Springs | Apache–Sitgreaves | AZ | 6 October 2017 | 2018 | 13 |
| Baca | Gila | NM | 12 May 2017 | 2018 | 23 |
| Bonita | Carson | NM | 3 June 2017 | 2018 | 27 |
| Boundary | Coconino | AZ | 1 June 2017 | 2018 | 14 |
| Flying R | Coronado | AZ | 14 June 2017 | 2018 | 15 |
| Frye | Coronado | AZ | 7 June 2017 | 2018 | 21 |
| Goodwin | Prescott | AZ | 24 June 2017 | 2018 | 11 |
| Hondito | Carson | NM | 16 May 2017 | 2018 | 7 |
| Kerr | Gila | NM | 1 May 2017 | 2018 | 14 |
| Lizard | Coronado | AZ | 7 June 2017 | 2018 | 9 |
| Pinal | Tonto | AZ | 8 May 2017 | 2018 | 9 |
| Rucker | Coronado | AZ | 7 June 2017 | 2018 | 9 |
| Sawmill | Coronado | AZ | 23 April 2017 | 2018 | 7 |
| Slim | Apache–Sitgreaves | AZ | 1 June 2017 | 2018 | 10 |
| Snake Ridge | Coconino | AZ | 19 May 2017 | 2018 | 20 |
| *Total* | | | | | *337* |

Tree measurements were collected on trees >10 cm diameter at breast height (dbh, 1.37 m) post-fire in each plot to characterize species, canopy cover, tree height, estimated pre-fire mortality, and fire-induced mortality [28]. We used the Central Rockies variant of the Forest Vegetation Simulator (11 January 2019 version) [29,30] to generate pre- and post-fire canopy cover and basal area estimates based on tree measurements and estimates of pre-fire mortality and fire-induced mortality, respectively. Fire-induced mortality was distinguished from pre-fire existing dead trees based on factors such as the amount of bark, depth of char, and presence of limbs and small branches similar to previous studies [31]. FVS uses established biometric equations that relate tree measurements to other tree metrics such as canopy cover and basal area [32,33]. The FVS includes a canopy cover adjustment factor (CCadj) based on the spacing of trees (five levels from random to uniform) to adjust

for overlapping tree crowns [32,34], which could be used to calibrate FVS-estimated canopy cover to actual conditions. We excluded plots with <10% pre-fire canopy cover to limit the dataset to forested lands.

### 2.3. Derivation of Satellite Imagery Indices

We derived burn severity indices from multi-spectral satellite imagery (the Landsat-8 Optical Line Imager (OLI) and Landsat-7 Enhanced Thematic Mapper Plus (ETM+) courtesy of the U.S. Geological Survey, and the Sentinel-2 Multispectral Imager (MSI) (Copernicus Sentinel data 2016–2018) [35]), each rescaled to top-of-atmosphere reflectance. In this paper, we refer to Landsat-7 ETM+ and Landsat-8 OLI collectively as "Landsat." We used OLI imagery except for a single case where ETM+ had clearer imagery. Consistent with the current RAVG workflow, the indices were calculated from a pair of satellite images— one each pre- and post-wildfire—judiciously selected by an analyst to reveal fire-related changes and minimize changes due to other factors such as annual productivity, seasonal phenology, or non-fire disturbances. To calculate the version of indices with the offset, an offset value was subtracted from the standard dNBR, and thus RdNBR and RBR equations, for each pair of indices to account for differences between pre- and post-fire images due to phenology [15].

Because GPS plot locations can be inaccurate, we smoothed satellite indices using adjacent pixels to account for potential location error. A 3-by-3 kernel weighting neighboring pixels based on the portion overlapped by a 60 m diameter circle was used to partially weight pixels which could overlap a 30 m diameter plot centered anywhere within 15 m of the center pixel's centroid (Figure 2, Tables 2 and 3).

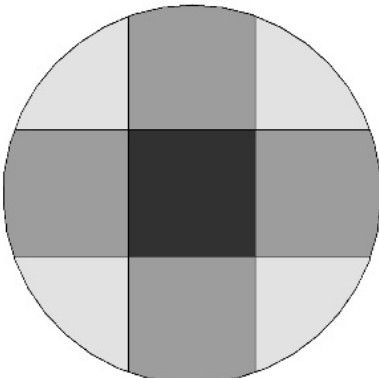

**Figure 2.** A schematic of the area of adjacent 20 m pixels within a 30 m radius circle factored into the weighting of the kernel used to smooth Sentinel-2 data.

**Table 2.** Kernel used to smooth 30 m Landsat data.

$$\begin{bmatrix} 0.025 & 0.146 & 0.025 \\ 0.146 & 0.320 & 0.146 \\ 0.025 & 0.146 & 0.025 \end{bmatrix}$$

**Table 3.** Kernel used to smooth 20 m Sentinel-2 data.

$$\begin{bmatrix} 0.0766 & 0.1377 & 0.0766 \\ 0.1377 & 0.1427 & 0.1377 \\ 0.0766 & 0.1377 & 0.0766 \end{bmatrix}$$

### 2.4. Photo-Interpretation Sampling

Because direct estimates of canopy cover from 20–30 m resolution satellite imagery are poor without proper training, canopy cover estimates were derived remotely from photographic interpretation of high-resolution (30 cm) imagery from the USDA Forest Service

Southwest Region photogrammetry program using a point intercept method for assigning canopy cover values (e.g., canopy/no canopy) to gridded points within a plot [36,37]. The PI data also increased the sample size to areas relatively inaccessible by field crews. Existing pre- and post-fire aerial resource photography was obtained for forests that burned in 2017 and 2018. Pre-fire aerial photos were limited to those acquired no more than five years prior to the given fire (Table 4) to prevent large differences in natural canopy cover change prior to the fire from influencing the data. The photo interpretation sampling area was cross-checked against insect and pathogen aerial detection surveys to confirm that none overlapped areas with extensive non-fire mortality events.

**Table 4.** Fires sampled, dates of fire, and pre- and post-fire aerial photos.

| Fire (National Forest) | State | Year of Fire | Year of Pre-Fire Aerial Photos | Year of Post-Fire Aerial Photos | Number of PI Plots (Number of OS Plots) |
|---|---|---|---|---|---|
| Tinder (Coconino) | AZ | 2018 | 2014 | 2018 | 35 (12) |
| Goodwin (Prescott) | AZ | 2017 | 2015 | 2017 | 13 (4) |
| Sardinas Canyon (Carson) | NM | 2018 | 2014 | 2018 | 33 (8) |
| Deiner (Cibola) | NM | 2018 | 2016 | 2018 | 29 (10) |
| Blue Water (Cibola) | NM | 2018 | 2016 | 2018 | 32 (10) |
| Pinal (Tonto) | AZ | 2017 | 2012 | 2017 | 23 (3) |
| Fires below not field sampled | | | | | |
| Highline (Tonto)/ Bears | AZ | 2017 | 2012 | 2017 | 19 |
| Redondo RX (Cibola) | NM | 2018 | 2016 | 2018 | 18 |
| *Total* | | | | | 202 (47) |

Photo interpretation plots were located using two systems. First, to characterize fire-induced changes across the entire burn perimeter, a systematic grid of 100 potential plots was generated for each of the 8 fires where pre- and post-fire aerial resource photography was available. Grid spacing was adjusted for each fire to retain up to 40 PI plots per fire after stratification and exclusion of plots due to low canopy cover or edge effects. The gridded photo plots were stratified evenly across immediate assessment RdNBR values [12]. Fire perimeters were buffered by −60 m, meaning only the area >60 m interior of the fire perimeter was sampled to avoid plots being partially in or out of the fire. Unburned photo plots were identified within the fire perimeter and also within a 500 m buffer outside of the fire perimeter to allow for approximately half of unburned plot sampling to occur outside the fire perimeter. Second, to relate remotely sensed data to field observations, another 47 photo plots were over-sampled (OS) coincident with a stratified random sample of field plots. Field-sampled plots were buffered by 250 m so that the two sets of plots did not overlap. Plots with <10% canopy cover pre-fire were omitted from all analyses.

A circular plot with a diameter of 40 m was used for PI. A PI technician attributed a grid of 37 points within each 40 m plot as live tree canopy, shrub canopy, or bare ground [37] using the Image Sampler, an add-on to Esri ArcMap that aids in sampling aerial photos. Where shadows or edges made attributing cover unreliable, points in question were discarded. If more than 20% of sample points were discarded, the entire PI plot was discarded. A combined 202 PI plots (grid and OS) were used in analysis.

*2.5. Accounting for Canopy Reduction due to Fire*

Canopy scorch and torch post-fire are inherently accounted for in the PI data; however, FVS has no direct mechanism to estimate losses in canopy cover due to fire. Miller et al. (2009) [17] used a reduction of tree crown footprint based on field-measured increases in crown base height to reduce tree crown widths, using crown volume shapes published for California. No crown volume equations are readily available for the Southwest. Additionally, in many instances, tree crown reduction due to fire does not occur uniformly from the

bottom up. To account for scorch post-fire in field plots, we utilized a two-step process. First, we subtracted the change in FVS cover (FVS Δ CC) from the change in PI cover (PI ΔCC) to adjust the FVS-modeled canopy cover change for scorch (FVS ΔCC − PI ΔCC = scorch adjustment). Second, we utilized *k*-nearest neighbor (KNN) regression to relate the scorch adjustment to CBI (scorch adjustment vs. CBI). This KNN coefficient was then used to increase FVS-generated canopy cover change to account for scorch. To test whether the change in canopy cover between the two methods was comparable and warranted combining data for overall model development, PI ΔCC and scorch-adjusted FVS ΔCC were compared to each other using a simple linear regression through the origin [38].

*2.6. Model Development*

To improve model accuracy based on anticipated difficulties in estimating burn severity in this region, we explored the use of additional variables and modeling methods beyond the current parametric models [17] used in RAVG fire severity and forest change products. Similar to previous studies, we included several variables derived from a 30 m DEM (U.S. Geological Survey, Reston, VA, USA) [39] shown to be relevant to burn severity, including elevation, slope, aspect, and topographic convergence index [31,40–43]. We evaluated non-parametric models because the utility of several of these algorithms has been demonstrated in the field of fire effects prediction, including random forest [40,41], boosted regression trees [42], and general additive models (GAM) [32]. Our response variables can be characterized as proportions which typically includes a mass of observations at 0 and 1 with continuous data between these bounds. Zero-and-one inflated data follow the beta (ZOIB) distribution [44], which we used in a GAM. We tested a variety of standard satellite indices that have been shown to predict burn severity and forest change, including RdNBR, dNBR, and RBR [12,13,16], and we tested each index with and without "offsets," values calculated in unburned areas near each fire and intended to account for non-disturbance differences between the pre- and post-fire images. Given the high potential for monsoonal rains to occur quickly following fire, resulting in loss of ash cover, we did not use the conversion factor derived by Miller and Quayle (2015) [45] to modify 1-year post-fire models to produce immediate post-fire estimates for burn severity and stand change. Instead, we developed stand-alone models for immediate post-fire effects based on image pairs that used post-fire imagery captured immediately after the fire.

To predict burn severity, canopy cover change, and basal area change after wildfire, we developed and tested a series of parametric and non-parametric models, with data obtained from the Southwest. To limit the overall number of models evaluated, model form and different predictor variables were each evaluated in turn, rather than evaluating every permutation of model form and predictors (Figure 3). Only the best candidate models were carried forward into the next evaluation, i.e., once the model form was chosen, satellite indices were evaluated on only the selected model form. To compare candidate model performance, test mean squared error (MSE) was computed within a 7-fold cross validation and then averaged across folds. The details of the models sequentially tested are given in Appendix A.

The parametric models fit were of the form currently used in RAVG production (Craig Baker, pers. comm.), which are the inverse of those documented in Miller et al. (2009) [17] and follow a sine curve for ΔBA and ΔCC and natural log curve for CBI (Equations (1)–(3)). The application of these parametric models (Miller et al., 2009) [17] includes limits below and above which predictions are set to the minimum and maximum values, that is, zero and 100% loss, respectively, for ΔBA and ΔCC and zero and three, respectively, for CBI.

$$\text{CBI} = \left(\frac{1}{c}\right)\ln\left(\frac{Index - a}{b}\right) \tag{1}$$

$$\Delta\text{BA} = \sin\left(\frac{Index - a}{b}\right) \tag{2}$$

$$\Delta\text{CC} = \sin\left(\frac{Index - a}{b}\right) \qquad (3)$$

*Index* refers to any one of the satellite indices tested (dNBR, RdNBR, or RBR) and *a*, *b*, and *c* are constants.

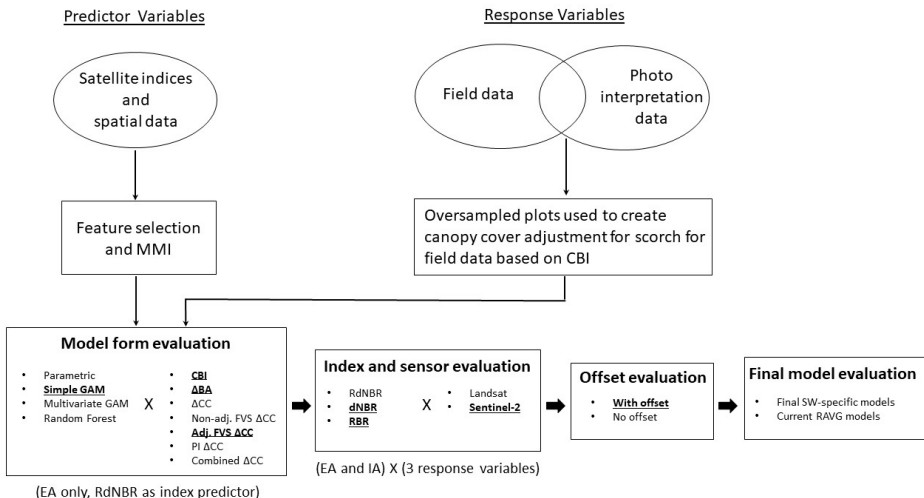

**Figure 3.** A flowchart of the analysis process. Circles indicate datasets and squares indicate analysis processes. Thin arrows depict where specific processed data are used; see Methods Section and Appendix A for complete description of all data used. Bold arrows indicate model evaluation steps taken in sequence. Variables and model form in bold and underlined were those carried forward into successive model development steps.

## 3. Results

### 3.1. Model Development Process

Initial model development produced extensive results, which can be found in Appendices A and B. Models built from just the field-derived canopy cover datasets predicted better than those using combined canopy cover data (field- and PI-derived), so only the field-derived canopy cover response variable was carried forward. General additive models (GAMs) performed best as a group and were the model form carried forward into final model development. Multivariate GAMs were evaluated, but due to mixed results and the complexity of acquiring additional variables in the production phase, we elected to only carry forward simple GAMs into the final models (Table A3). Indices derived from Sentinel data performed better than indices derived from LANDSAT data. The indices that performed best and which were carried forward into final model development were the RBR for the EA time series and dNBR for the IA (Tables A6 and A7). Indices with the offset used in calculations generally performed the best, so all final models were built using indices with the offset included (Table A8).

### 3.2. Final Models

We compared the best candidate models (simple GAMs) to those used in the current RAVG products as of this writing [17]. The models built from Southwest data all had a lower test MSE than the current RAVG models [17] when applied to the Southwest data, suggesting that the Southwest models perform better when viewed as continuous data products. However, the results for accuracy and Kappa, which are used to evaluate categorical data response, were mixed (Table 5). For CBI, all three metrics (test MSE, accuracy, and Kappa) suggested that the models built from Southwest data perform better in the Southwest. Conversely, the Miller et al. (2009) [17] canopy cover change models had higher accuracies and Kappa, though differences were mostly small. Likewise, for ΔBA the current RAVG [17] models had higher accuracy and Kappa (Table 5). However, for

BA change for both EA and IA timeframes, several datapoints in the highest burn severity categories were predicted into the lowest burn severity category and vice versa with the current RAVG models [17] (Tables A9–A17). This does not diminish accuracy, but the magnitude of the error possible is higher than in the models built using the Southwest data. The model accuracy metric only factors in a binary response at each category and does not penalize differently for large versus small categorization errors. These results suggest that the ΔBA and ΔCC models built from Southwest data predict better as continuous products for the Southwest, but current RAVG models [17] predict the most observations in the correct categories.

**Table 5.** Comparison of our final SW models predicting CBI, ΔBA, and ΔCC (percent accuracy, Kappa, and test MSE) to models currently used in RAVG burn severity and forest change products at the time of this writing [17].

| | IA SW-Specific Model (Sentinel-2 dNBR) | | | IA Current Model (Landsat RdNBR) | | | EA SW-Specific Model (Sentinel-2 RBR) | | | EA Current Model (Landsat RdNBR) | | |
|---|---|---|---|---|---|---|---|---|---|---|---|---|
| | Acc. | Kappa | Test MSE | Acc. | Kappa | Test MSE | Acc. | Kappa | Test MSE | Acc. | Kappa | Test MSE |
| CBI | 61.4 | 46.7 | 0.0184 | 46.9 | 32.1 | 0.8753 | 62.0 | 47.0 | 0.0237 | 50.1 | 35.9 | 1.1265 |
| ΔBA | 52.2 | 33.9 | 0.0409 | 67.4 | 47.5 | 0.0547 | 56.1 | 38.1 | 0.0407 | 63.8 | 46.9 | 0.0705 |
| ΔCC | 50.7 | 38.0 | 0.0347 | 54.9 | 41.5 | 0.0886 | 53.4 | 41.2 | 0.0337 | 54.9 | 41.4 | 0.0518 |

## 4. Discussion

Our objective was to create and explore the efficacy of region-specific models for the southwest U.S. predicting burn severity and forest change with fire. Surprisingly, we found that current RAVG models are better categorical predictors for forest change than our region-specific models, although our models appear to serve as better continuous prediction tools. Our methods (Appendix A) and publicly available code (https://github.com/alreiner/SW_RAVG.git) (accessed on 1 August 2022) provide for the development of similar region-specific models in other systems.

### 4.1. Efficacy of Region-Specific Models in Assessing Post-Wildfire Change

We assessed the performance of region-specific fire effects models with test metrics for assessing the model's ability to fit a continuous response or categorize the response. Categorical prediction is best assessed with the confusion matrix and accuracy, whereas test MSE is a metric more suitable for evaluating a continuous response. In general, the Miller et al. (2009) [17] ΔBA and ΔCC models may have better categorical prediction capabilities than models developed from the SW data; however, continuous and overall predictions from the Southwest models are superior when applied to Southwest data (Table 5). When the total vegetation cover is low in a spectral image, changes to the vegetation have lower impact on the spectral image, which could make for a weaker relationship between the indices and fire effects. Effectively, the sparser vegetation has the effect of increasing the substrate signal, which influences the indices. A wide variety of forested vegetation types are present in the Southwest and in our dataset, ranging from pinyon–juniper woodland to mixed conifer forest. The variation in the spectral signature may be wider than that of Miller et al. (2009) [17], which did not include arid woodlands.

Nuances between satellite indices factor into why various indices performed better in Southwest models. Cansler and McKenzie (2012) [46] note that in areas with little variation in pre-fire reflectance, meaning homogenous vegetation cover, dNBR has little advantage over RdNBR. Parks et al. (2014) [15] note that areas with very low pre-fire NBR can cause very high or very low RdNBR due to the square root in the denominator, leading RBR to potentially perform better. The Southwest has highly variable canopy cover and many vegetation types with low cover, so it is not surprising that RBR proved optimal in some instances. A few factors could explain why dNBR was the best IA predictor, whereas RBR was the best EA predictor. Parks et al. (2014) [15] note that dNBR is correlated to pre-fire NBR. Severity is understandably correlated to pre-fire vegetation cover in the Southwest,

as stand-replacing fire regimes occur in the highest elevation sites dominated by mesic forests, which inherently have high vegetation cover compared to the arid woodlands of the low elevations. For the IA time series, this correlation would likely boost the predictive capacity of a satellite index. However, for the EA time series, derived one growing season after the fire, giving the understory more time to recover, RBR, an index normalized by pre-fire vegetation cover, was favored, suggesting that normalizing the index to pre-fire vegetation cover is more useful for modeling at that timeframe. This normalization makes finer differences in dNBR more apparent in the lower-cover portions of the study area, which likely had more understory recovery than the closed-canopy forest types. The Sentinel-2-based indices may have performed better than the Landsat indices partially due to the finer scale (20 m rather than 30 m) [47].

The zero-and-one inflated beta distribution [44] is suitable where a binary response is a frequent outcome in an otherwise continuous data distribution. In the context of fire severity, ΔCC, and ΔBA, these are unburned plots or plots with 100% canopy scorch. Accounting for this distribution in a GAM may have given the GAM models an advantage over the parametric and random forest models by better addressing the binary nature of the data. The Southwest GAM models generally outperformed the Southwest parametric models. It is plausible that GAMs using a zero-and-one inflated beta distribution derived with the Miller et al. (2009) [17] data might predict with greater accuracy.

In our analyses, multivariate GAM performance may have been reduced by several factors. The stepwise procedure we utilized for multivariate model development is reliant on appropriate selection of plausible a priori variables with potential collinearity between variables. For this reason, it can be sensitive to over-fitting if care is not taken when selecting a priori variables [48]. However, we used a limited selection of a priori variables and selected only those with sufficiently high importance. The gamlss package available in R allows the use of the zero-and-one inflated beta distribution; however, it does not incorporate model selection algorithms utilizing shrinkage such as LASSO. The shrinkage algorithms would be more effective at reducing the moderate to low importance predictors to zero. In our analyses, the relationships of the non-satellite predictors were weak, so there was little additional information to be added from each. Holden et al. (2009) [40] noted that topographic variables which describe moisture availability present shifting, and perhaps conflicting, roles with increasing elevation. For example, at lower elevations, aspect can influence vegetation distributions and fuel loads due to changes in moisture availability. It is possible that at lower elevations, only northerly aspects have enough moisture and fuels to burn with high severity. Conversely, at higher elevation where mixed conifer forests are dominant, southerly aspects or areas that experience lower moisture have adequate fuels to burn with high severity and could be more likely to do so given the lower fuel moistures. These differences in the way aspect and other geomorphic predictor variables relate to severity with increasing elevation may be important for process-based modeling but are less useful for multivariate GAMs. Models produced from larger datasets and machine learning methods capable of incorporating complex interactions may capture these interconnections better.

### 4.2. Influence of Forest Change Measurements on Error

Field measurements and photo interpretation methods each have an irreducible error when estimating canopy cover, which can weaken canopy cover models. Error in field methods can arise from measurement or sampling error, and error can be introduced in photo interpretation due to shadows or edges being less interpretable. Field plots were intentionally located in areas of relatively homogenous severity [17], whereas PI plots were located on a systematic grid, which could have increased the ratio of PI plots in areas of mixed severity, potentially muting the CBI to satellite index relationship.

Changes in canopy cover and tree mortality in areas of heterogeneous severity would have a weaker relationship between predictor and response variables. This may have led to the PI models having weaker relationships with satellite indices when compared to

the field plots. Background mortality may also be a confounding error source in photo interpretation where pre-fire photos were taken several years before the fire and some background mortality would be expected even in the absence of fire. Similarly, in this analysis, the post-fire aerial resource photography utilized was collected the same year as the fire, which would not pick up delayed mortality for the EA timeframe. Field data were collected the year after fire, so include the 1-year delayed mortality with the assumption of EA mortality being the same as IA, which may not be entirely accurate but is likely a minor error considering overall error sources and precision of the data and models.

### 4.3. Implications and Directions for Future Research

The implications of this research support continued development of the synergy between remotely sensed data and machine learning methods as well as appraisal of current models and development of improved or region-specific models. With the increasing variety of remotely sensed data as well as machine learning methods with which to develop models, more nuance is possible in fire effects modeling. New sensors are being developed each year, expanding the capacity for remote data collection, and data post-processing methods as well as machine learning algorithms are continually being expanded and improved. New sensors and machine learning will aid in more precise and accurate model development in the years to come. Automated workflows could improve the use of these data and machine learning methods. The improvements and lessons our Southwest models offer include addressing the zero-and-one inflated beta distribution inherently with the choice of model form and algorithm. A drawback to the parametric models currently used in RAVG products is the need to apply limits to the sine and natural log functions at points of inflection or nonsensical predictions, capping them at minimum and maximum predictions. These limits affect the categorization of a portion of the data range, which can influence categorization errors. Our Southwestern-specific models and approach to other arid or semi-arid regions, such as the southwestern Rockies and the Great Basin, may improve fire effects modeling and provide better information to researchers and managers under increasingly variable fire regimes.

Future research could benefit from three tactics not employed in this project: using the individual bands from the remote satellite sensors rather than indices, exploring additional machine learning methods other than GAM with modifications to accommodate the zero-and-one inflated distribution, and employing composite images through Google Earth Engine (GEE). Indices are useful in that they compile information from several relevant variables into one variable, making models, relationships, and predictions easier to understand. However, there is some information loss when multiple variables are combined into an index. Applying multivariate and machine learning modeling methods to the variables addressed in this research plus the individual band differences or individual bands such as pre- and post-fire bands five and seven and NBR may provide additional predictive power [49]. Additional variables not used in this study could improve model results, namely active fire data such as those derived from MODIS and VIIRS [50,51]. This concept could be taken a step further by exploring linear unmixing, in which the entire spectrum of image information is used rather than categorizing the image data into bands [52]. The gamlss package in R is one of the few ready-made algorithms available to model the binary and continuous response of a dataset simultaneously. It is possible to split data into binary and continuous response subsets and model them separately; however, as the split is non-random, these models will be applied to data on which they were not developed, which results in sample selection bias. Methods and algorithms to overcome this bias are being developed and should be explored to allow a variety of proven machine learning methods beyond GAM to be applied. Previous studies [42] have used the GEE environment to create composite images from collections of imagery based on date and quality constraints, from which more robust models can be developed as these data moderate differences in individual images. Given that we focused on exploring Sentinel-2 data just as data from this sensor were becoming available, we did not have a broad history

from which to pull multiple images for the 2017 fires sampled, so we opted not to pursue the use of composite images. However, others have found value in this approach [42], which warrants future exploration.

Appropriately designed field training data are key building blocks to creating improved or regional models. Although CBI has historically been utilized as the primary response variable in burn severity models, there is value in collecting and modeling as response variables other more mechanistically linked or forest structure data rather than unitless and subjective data such as CBI [50,53]. Additionally, field verification of new and existing models could help to highlight areas where revision to current models would be beneficial. Archiving data and methods would greatly facilitate re-analysis of historical datasets with more contemporary statistical learning tools as well as meta-analyses using combined data or the use of similar datasets as validation sets for model development.

## 5. Conclusions

Our region-specific post-wildfire model had several advantages over conventional California-based models and showcases the utility of developing region-specific models. However, measurement error, limitations to current statistical packages, and the complexity of untangling the remote-sensing data spectrum are among the potential issues with developing and implementing similar models to assess post-wildfire change. Continued development, collection, and archival of remote and ground-based data will provide for better calibration and more accurate decision making. Our success in improving on the Miller (2009) [17] models should provide guidance for future region-specific adaptations of these models.

**Author Contributions:** Conceptualization, A.L.R., M.W. and C.B.; field data collection, A.L.R., M.W., C.B. and Enterprise Program field technicians; remote-sensing index data, C.B.; analysis, A.L.R.; analysis review and revision, J.D.B., B.M.R., M.W. and C.B.; results interpretation, A.L.R., C.B., M.W., B.M.R. and J.D.B.; writing, A.L.R.; review and editing, A.L.R., C.B., M.W., B.M.R. and J.D.B. All authors have read and agreed to the published version of the manuscript.

**Funding:** This project was funded by the USDA Forest Service Geospatial and Technology Applications Center.

**Informed Consent Statement:** Not applicable.

**Data Availability Statement:** Data can be assessed from the USDA Forest Service Research Data Archive: field and plot data, including coordinates, are located at https://doi.org/10.2737/RDS-2022-0018 (accessed on 12 September 2022), and satellite indices created from image pairs along with fire perimeters are located at https://doi.org/10.2737/RDS-2022-0019 (accessed on 12 September 2022). Code is available on Github at (https://github.com/alreiner/SW_RAVG.git) (accessed on 9 September 2022).

**Acknowledgments:** We are grateful for the collaboration between the USDA Forest Service Geospatial Technology and Applications Center and the Enterprise Program for making this project possible. We would like to thank Brian Harvey and Saba Saberi for sharing field and analysis methods for a similar modeling effort in the northwest U.S. We also thank Sara Levy for coordinating field operations, as well as all the efforts of the various crew members. We appreciate the thoughtful review and suggestions provided by Andy Hudak, USDA Forest Service, Rocky Mountain Research Station, Moscow, ID, which greatly improved this manuscript. We are grateful to the U.S. Geological Survey and the European Space Agency for making Landsat and Sentinel-2 images freely available. We are grateful for the R Project as well as the developers of statistical and geospatial data processing packages.

**Conflicts of Interest:** The authors declare no conflict of interest.

## Appendix A

*Appendix A.1. Model Development Methods and Intermediate Results*

Appendix A.1.1. Model Evaluation Metrics and Feature Selection

We chose to use test MSE averaged across a sevenfold cross validation to compare models, although a variety of test metrics have been used in similar studies to compare candidate models [54]. Other metrics include accuracy and Kappa, generated from confusion matrices, and area under the receiver operating curve (AUC). Overall accuracy, defined as the "degree of right predictions of a model" [54], and Kappa (or Cohen's Kappa), which is the difference of the overall accuracy of the model and that of pure chance [55], are commonly used in assessing geospatial mapping accuracy. However, accuracy can give an overly optimistic score for models with heavy class imbalance [56], and Kappa also has drawbacks because it is a relative score that is also affected by unbalanced categories [57]. The area under the receiver operating curve (AUC) is a measure of the ability of a (binary) classifier to distinguish between classes and has been used in the assessment of severity classification models [16]. However, AUC is typically applied to classification and can involve dichotomizing a non-binary response [31]. We chose to use test MSE to test candidate models because it describes the deviation of model predictions from training data and does not utilize common, yet arbitrary, classes. We show accuracy and Kappa for final models for comparison to previous studies. Accuracies were computed on the same model-development dataset for a direct comparison of final (whole dataset) models [42] to current models [17].

For multivariate models tested, feature selection using correlations and multi-model inference approach (MMI) was completed to reduce the number of available predictors to a smaller set. Correlation coefficients were used in feature selection to reduce redundant and marginally useful predictors [58]. The Kendall's tau correlation coefficient was used due to the non-linear relationships between predictors and response variables, as well as the lack of normality in the distributions for most variables [59,60]. We used the MuMin v4.0.5 [61] package to compare all possible combinations of plausible variables and rank models by second-order Akaike information criterion (AICc). The relative variable importance (RVI) was computed for each variable, and variables with RVI > 0.5 were considered important [48] and brought forward in multivariate model development.

Appendix A.1.2. Non-Parametric Modelling Methods

We fit two types of non-parametric models: a random forest and a general additive model (GAM). A random forest is a multivariate learning algorithm that combines many decision trees into a final model outcome [62]. A GAM is an additive model that uses smoothing to accommodate potentially nonlinear relationships for individual predictor variables. Random forests have the advantage of modeling complex interactions of covariates, but they lack interpretability, whereas GAMs are more interpretable and can model nonlinear and "hockey-stick" relationships. We used the randomForest v4.6-14 package (R documentation, randomForest v4.6-14) [63] to fit a random forest model with the topographic variables explored during feature selection, plus Landsat-derived pre-fire NBR and EA RdNBR (Table A5). We used the gamlss package version 5.3-2 in R to fit a GAM using a zero-and-one inflated beta (ZOIB) distribution [64,65]. The stepGAIC function in the gamlss package was used to determine multi-variate GAM formulas for each parameter [64]. In addition to the topographic variables selected during the feature selection phase, we added pre-fire NBR as a potential variable for the stepGAIC function when finding the optimal multivariate model, to aid the satellite indices as predictors in sparse vegetation types. The parameters modeled by gamlss for the ZOIB distribution (family = BEINF) allow for prediction of the continuous nature of the data (mu), as well as the probability of zero (nu) and one (tau) (R documentation, gamlss version 5.3-2). Mu, nu, and tau were combined to generate a continuous response, which factors the probability of

zero and one (Equations (A1) and (A2)) in with the continuous response (Equation (A3)) (pers. comm., Saba Saberi):

$$p0 = \mathrm{nu}/(1 + \mathrm{nu} + \mathrm{tau}) \tag{A1}$$

$$p1 = \mathrm{tau}/(1 + \mathrm{nu} + \mathrm{tau}) \tag{A2}$$

$$Yest = (1 - p0) * (p1 + (1 - p1) * \mathrm{mu}) \tag{A3}$$

where $p0$ and $p1$ are the probability at 0 and 1, respectively, and Yest is the predicted response.

Appendix A.1.3. Canopy Cover Estimation Results

Pre- and post-fire canopy cover estimates from FVS and PI methods were compared. The FVS utilizes categorical classifications of tree spacing to adjust canopy cover due to tree canopy overlap. For pre-fire data, the "Very Uniform" canopy cover adjustment (CCadj) in FVS yielded the best match between the FVS and the PI data; therefore, the "Very Uniform" CCadj was used in FVS to generate canopy cover for all field data. For post-fire data, the "Somewhat Uniform" or "Moderately Uniform" yielded the best match, illustrating that FVS-generated canopy cover requires an adjustment to account for green tree foliage removed by fire through needle scorch and torch (Table A1).

**Table A1.** Summary statistics for canopy cover (along with 5 different levels of canopy cover adjustment factor for FVS-generated values ranging from "random" tree spacing to "extremely uniform") for the over-sampled plots.

| Method | Min | 1st Quartile | Median | Mean | 3d Quartile | Max | Standard Deviation |
|---|---|---|---|---|---|---|---|
| Pre-Fire | | | | | | | |
| FVS (Extremely Uniform) | 38.7 | 80.3 | 87.7 | 84.6 | 92.6 | 100.0 | 11.5 |
| FVS (Very Uniform) | 24.0 | 59.7 | 69.1 | 67.2 | 76.7 | 99.2 | 14.2 |
| FVS (Moderately Uniform) | 17.9 | 48.0 | 57.1 | 56.1 | 65.0 | 96.8 | 14.3 |
| FVS (Somewhat Uniform) | 13.9 | 39.1 | 47.3 | 47.0 | 54.8 | 92.6 | 13.7 |
| FVS (Random) | 12.2 | 35.0 | 42.7 | 42.6 | 49.9 | 89.5 | 13.2 |
| Photo interpretation (PI) | 22.0 | 57.0 | 70.0 | 67.6 | 79.5 | 100.0 | 18.9 |
| Post-Fire | | | | | | | |
| FVS (Extremely Uniform) | 0 | 45.2 | 75.2 | 62.3 | 86.7 | 100.0 | 33.4 |
| FVS (Very Uniform) | 0 | 28.8 | 54.2 | 47.4 | 67.8 | 99.2 | 27.5 |
| FVS (Moderately Uniform) | 0 | 21.7 | 43.0 | 38.8 | 55.7 | 96.8 | 23.6 |
| FVS (Somewhat Uniform) | 0 | 17.0 | 34.6 | 32.0 | 46.0 | 92.6 | 20.3 |
| FVS (Random) | 0 | 14.9 | 30.9 | 28.9 | 41.5 | 89.5 | 18.7 |
| Photo interpretation (PI) | 0 | 11.0 | 35.0 | 36.3 | 59.0 | 92.0 | 27.6 |

To calibrate the FVS-modeled change in canopy cover to include partial tree crown reduction due to scorch and torch (in addition to tree mortality), we utilized KNN regression between PI-adjusted canopy cover change and the tree portion of the CBI (i.e., the CBI components attributed to the upper two strata). The resultant coefficient had a maximum of −0.14 and a minimum of −0.25. The KNN-predicted difference is consistent with how scorching patterns may affect the canopy, in that little modification occurs at low severity, a fair amount of modification occurs at moderate severity, and at high severity, where many trees are completely torched and therefore not considered in FVS calculations, less modification is needed (Figure A1).

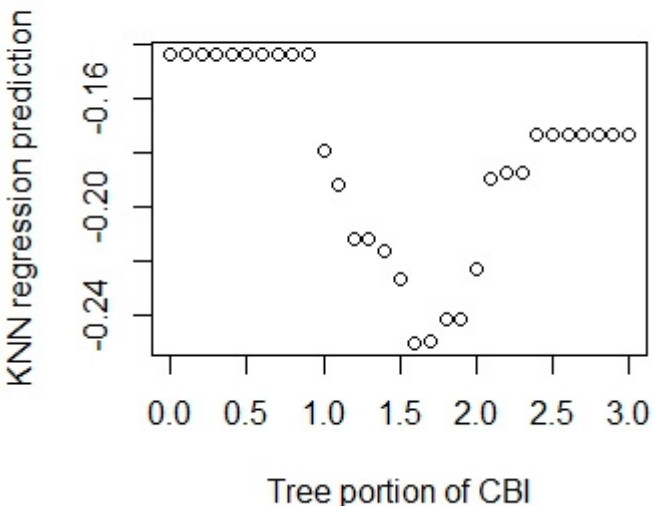

**Figure A1.** KNN regression predicted difference of FVS-generated canopy cover change minus PI-generated canopy cover change versus tree portion of CBI.

The KNN-derived coefficient was applied to the FVS-generated canopy cover change estimates to account for tree canopy removed by fire due to scorch and torch. The adjustment was not applied to values where the tree portion of CBI was less than 0.5, because those low burn severity plots would be expected to show minor to low canopy cover loss. Canopy cover change was capped at 100%. A simple linear no-intercept regression between the adjusted FVS canopy cover change and the PI canopy cover change had an adjusted r-squared of 95.81%, demonstrating a strong relationship between the two methods (Figure A2). Therefore, the combined canopy cover change dataset was carried forward into analysis as a potential response variable ("combined ΔCC") in addition to the separate FVS and PI ΔCC datasets.

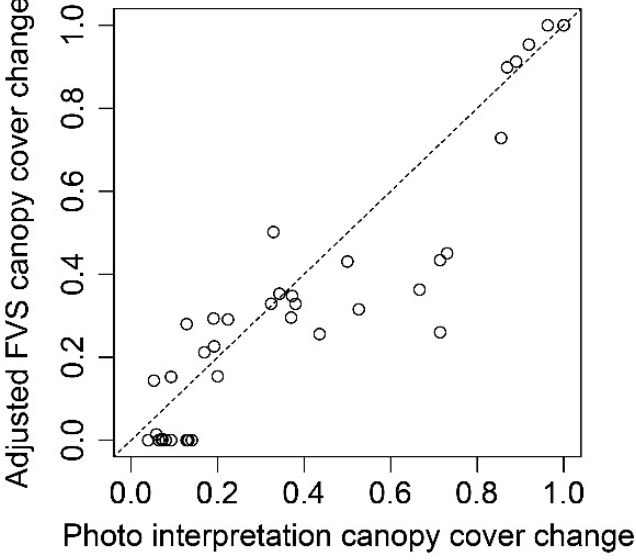

**Figure A2.** Photo interpretation canopy cover change versus scorch- and torch-adjusted FVS-generated canopy cover change and linear regression line.

Our methods of adjusting FVS-derived canopy cover from field measurements with the KNN regression with a limit at 0.5 for the tree portion of CBI created a cluster of scorch-adjusted canopy cover datapoints at 0.14. Further use of these data as input to other modeling products will be affected by this artificial and uneven data distribution. In raster data, this data cluster at 0.14 could be moderated by smoothing.

Appendix A.1.4. Feature Selection Results

The topographic and ecological predictor variables were explored to remove redundant variables and include those variables that would provide the most information to the models. The three response variables at the bottom of the matrix (Figure A3) are the composite burn index (CBI), change in canopy cover with fire (ΔCC), and change in basal area with fire (ΔBA). A first cut at feature selection using Kendall's tau correlations indicated correlations between the model response variables (ΔBA, ΔCC, and CBI) and topographic variables slope, elevation, and TCI. LANDFIRE Biophysical Setting (BPS) code had low correlation to change in basal area [66].

Multi-model inference (MMI) was then performed using the top four predictor variables (slope, elevation, TCI, and LANDFIRE BPS code) that showed measurable correlation with response variables (ΔBA, ΔCC, and CBI) along with the satellite index (RdNBR) used in previous RAVG models. The RdNBR with offset from Landsat Extended Assessment (EA) data was the most highly ranked predictor variable for all response variables based on relative variable importance (RVI; Table A2). Elevation was the second most important predictor for the model predicting CBI, whereas TCI was the second most important predictor for the other response variables (Table A2). The LANDFIRE BPS code was correlated to ΔCC, but not to CBI or ΔBA, and therefore was not included (Table A2). Due to the corroboration between Kendall's tau correlations and MMI results for TCI, elevation, and slope, these three predictor variables were carried forward in GAM multivariate model development.

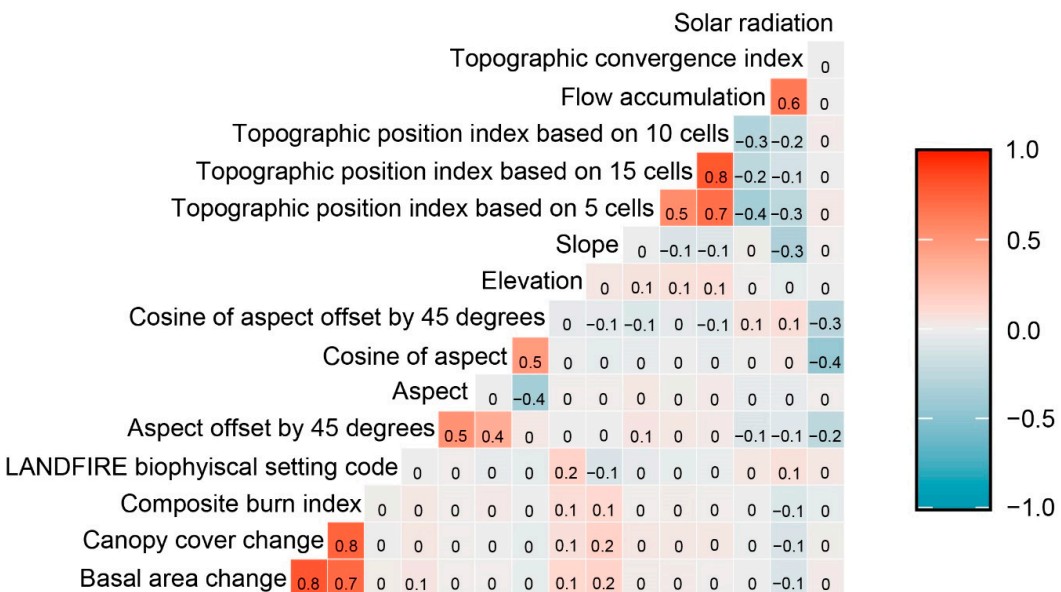

**Figure A3.** Kendall's tau correlations between response variables (CBI, ΔCC, and ΔBA) and topographic and ecological variables.

**Table A2.** Relative variable importance for the "best" models for each response variable. Predictor variables with RVI ≥ 0.50 are in bold.

| Response Variable | Relative (Predictor) Variable Importance for the "Best" Models | | | | |
| | RdNBR *** | Elevation | TCI | Slope | BPS Code |
|---|---|---|---|---|---|
| CBI | **1** | **0.66** | 0.44 | 0.35 | |
| ΔBA | **1** | 0.44 | **0.50** | 0.33 | |
| FVS ΔCC * | **1** | 0.43 | **0.50** | 0.36 | |
| PI ΔCC ** | **1** | 0.38 | **0.79** | 0.39 | 0.04 |

* FVS-generated canopy cover change was only computed for field dataset. ** PI-generated canopy cover change was only computed for PI dataset. *** RdNBR value used was generated from Landsat EA data with offset.

Appendix A.1.5. Model Form Evaluation Results

Several model forms as well as the different representations of the canopy cover response variable were compared using test MSE. Among the canopy cover change response variables, the scorch-adjusted FVS canopy cover change (adj. FVS ΔCC) had the lowest test MSE for every model form tested, with the parametric model having the lowest test MSE and multivariate GAM having the second lowest. Although the simple regression results indicated that combining the scorch-adjusted FVS and PI datasets would be warranted, the model created from that combined dataset did not have the lowest test MSE and so was not carried forward in analysis (Table A3). The scorch-adjusted FVS-derived canopy cover change data were the only ΔCC method carried forward in analysis after this point because it had lower test MSE across all model forms (Table A3). General additive models (GAMs) had the lowest test MSE for the CBI response variable, with multivariate GAMs being the second lowest. (Multivariate GAM model statements are below the variable dictionary in Table A4.) For ΔBA, the multivariate GAM had the lowest test MSE, and the simple GAM had the second lowest test MSE. The random forest model results generally had the higher test MSE. Based on these results, the simple GAM models were carried forward (rather than parametric or random forest) as the model form for comparisons of predictor variables.

**Table A3.** Comparison of test MSE (and standard deviation of test MSE in parentheses) averaged across 7 folds for each model form (parametric (Equations (1)–(3)), simple GAM, multivariate GAM, and Random Forest) using EA Landsat RdNBR with offset as a predictor.

|  | **Parametric** | **Simple GAM** | **Multivariate GAM** | **Random Forest** |
|---|---|---|---|---|
| CBI | 0.2486(0.0289) | 0.0273(0.0032) | 0.0267(0.0033) | 0.2657(0.0247) |
| ΔBA | 0.0504(0.0085) | 0.0483(0.0104) | 0.0473(0.0101) | 0.0518(0.0040) |
| Non-adj. FVS ΔCC | 0.0511(0.0081) | 0.0484(0.0102) | 0.0474(0.0094) | 0.0518(0.0040) |
| Adj. FVS ΔCC | 0.0392(0.0065) | 0.0397(0.0072) | 0.0390(0.0070) | 0.0440(0.0048) |
| PI ΔCC | 0.0514(0.0028) | 0.0523(0.0035) | 0.0530(0.0038) | 0.0607(0.0050) |
| Combined ΔCC | 0.0457(0.0015) | 0.0481(0.0041) | 0.0724(0.0032) | 0.0492(0.0027) |

Non-adj. FVS ΔCC is FVS-derived canopy cover change not adjusted for scorch. Adj. FVS ΔCC is scorch-adjusted FVS canopy cover change. PI ΔCC is the canopy cover change derived from the PI dataset. Combined ΔCC is canopy cover change derived from the combined scorch-adjusted FVS as well as PI canopy cover change.

**Table A4.** Dictionary of variables used in multivariate GAMs.

| **Variable Name** | **Data** |
|---|---|
| L_EA_rdnbr_with | EA Landsat RdNBR with offset |
| LEA_preN_f | EA Landsat pre-fire NBR |
| elev | Elevation |
| slope | Slope |
| TCI | Topographic convergence index |
| CBI.B | Overall CBI rescaled to 0-1 |
| pdBA | Pre- to post-fire percent change in BA |
| pdFVSVU | Pre- to post-fire percent change in non-scorch-adjusted FVS canopy cover |
| adj.lim.pdFVSVU | Pre- to post-fire percent change in scorch-adjusted FVS canopy cover |
| pdTreeCCloss | Pre- to post-fire percent change in PI-derived canopy cover change |
| pdCC | adj.lim.pdFVSVU and pdTreeCCloss datasets combined |

Multivariate GAM models, using EA timeframe Landsat RdNBR:.

Model: CBI
Model statement: gamlss(formula = CBI.B ~ L_EA_rdnbr_with + pb(TCI) + pb(L_EA_rdnbr_with), sigma.formula ≅ L_EA_rdnbr_with, nu.formula ≅ L_EA_rdnbr_with, tau.formula ≅ L_EA_rdnbr_with, family = BEINF, data = na.omit(SWRAVG_field_train))
Model: ΔBA

Model statement: gamlss(formula = pdBA~L_EA_rdnbr_with, sigma.formula ≅ L_EA_rdnbr_with + slope, nu.formula ≅ L_EA_rdnbr_with + LEA_preN_f, tau.formula ≅ L_EA_rdnbr_with + slope, family = BEINF, data = na.omit(SWRAVG_field_train))

Model: pdFVSVU

Model statement: gamlss(formula = pdFVSVU ~ L_EA_rdnbr_with + cs(elev), sigma.formula ≅ L_EA_rdnbr_with, nu.formula ≅ L_EA_rdnbr_with + TCI, tau.formula ≅ L_EA_rdnbr_with + elev + slope + LEA_preN_f + TCI, family = BEINF, data = na.omit (SWRAVG_field_train))

Model: adj.lim.pdFVSVU

Model statement: gamlss(formula = adj.lim.pdFVSVU ~ L_EA_rdnbr_with, sigma.formula ≅ L_EA_rdnbr_with + TCI, nu.formula ≅ L_EA_rdnbr_with + elev + LEA_preN_f, tau.formula ≅ L_EA_rdnbr_with + slope, family = BEINF, data = na.omit(SWRAVG_field_train))

Model: pdTreeCCloss

Model statement: gamlss(formula = pdTreeCCloss ~ L_EA_rdnbr_with + cs(slope), sigma.formula ≅ L_EA_rdnbr_with, nu.formula ≅ 1, tau.formula ≅ L_EA_rdnbr_with + slope, family = BEINF, data = na.omit(SWRAVG_PI_train))

Model: pdCC

Model statement: gamlss(formula = pdCC ~ L_EA_rdnbr_with + cs(elev) + cs(TCI), family = BEINF, data = na.omit(SWRAVG_train))

**Table A5.** Dictionary of variables used in random forest.

| Variable Name | Data |
| --- | --- |
| L_EA_rdnbr_with | EA Landsat RdNBR with offset |
| pdBA | Pre- to post-fire percent change in BA |
| BPScode | Landfire Biophysical Setting Code |
| asp_N45 | Aspect shifted to the north by 45 degrees |
| aspect | Aspect |
| cos_aspect | Cosine of aspect |
| cosasp_N45 | Cosine of aspect shifted to the north by 45 degrees |
| Elev | Elevation |
| slope | Slope |
| TPI_5cell | Topographic position index calculated across 5 cells |
| TPI_10cell | Topographic position index calculated across 10 cells |
| TPI_15cell | Topographic position index calculated across 15 cells |
| FlowAcc | Flow accumulation intermediate calculation from TPI |
| SolarRad | Solar radiation |
| TCI | Topographic convergence index |
| LEA_preN_f | EA Landsat pre-fire NBR |

Appendix A.1.6. Index and Sensor Evaluation

Comparisons were also made between results using different sensors (Landsat and Sentinel-2) and indices (RdNBR, dNBR, and RBR). Comparisons at this stage were made using simple GAMs for more direct comparison between different indices, although multivariate GAMs sometimes outperformed the simple GAMs. GAMs using indices from the Sentinel-2 sensors typically had lower test MSE than their Landsat counterparts (Tables A6 and A7). For both the Extended Assessment (EA) and Initial Assessment (IA) time series, the models with the lowest test MSE were formed from Sentinel-2 indices (Tables A6 and A7). For the EA time series, models using RBR had the lowest test MSE for all three response variables (Table A6), and IA time series models utilizing dNBR had the lowest test MSE for all response variables (Table A7); therefore, these indices were carried forward in model development.

**Table A6.** Test MSE (and standard deviation of test MSE in parentheses) averaged across 7-fold CV of single-predictor GAM models for the EA time series.

| | RdNBR | | dNBR | | RBR | |
| --- | --- | --- | --- | --- | --- | --- |
| | **Landsat** | **Sentinel-2** | **Landsat** | **Sentinel-2** | **Landsat** | **Sentinel-2** |
| CBI | 0.0273 (0.0032) | 0.0275 (0.0033) | 0.0260 (0.0021) | 0.0256 (0.0018) | 0.0244 (0.0020) | 0.0240 (0.0018) |
| ΔBA | 0.0483 (0.0104) | 0.0455 (0.0087) | 0.0495 (0.0062) | 0.0465 (0.0052) | 0.0450 (0.0070) | 0.0416 (0.0059) |
| Adj. FVS ΔCC | 0.0397 (0.0072) | 0.0363 (0.0056) | 0.0412 (0.0042) | 0.0378 (0.0032) | 0.0378 (0.0048) | 0.0340 (0.0039) |

**Table A7.** Test MSE (and standard deviation of test MSE in parentheses) averaged across 7-fold CV of single-predictor GAM models for the IA time series.

| | RdNBR | | dNBR | | RBR | |
| --- | --- | --- | --- | --- | --- | --- |
| | **Landsat** | **Sentinel-2** | **Landsat** | **Sentinel-2** | **Landsat** | **Sentinel-2** |
| CBI | 0.0333 (0.0022) | 0.0273 (0.0018) | 0.0215 (0.0021) | 0.0185 (0.0020) | 0.0227 (0.0021) | 0.0193 (0.0021) |
| ΔBA | 0.0710 (0.0092) | 0.0642 (0.0103) | 0.0440 (0.0052) | 0.0416 (0.0069) | 0.0454 (0.0057) | 0.0424 (0.0079) |
| Adj. FVS ΔCC | 0.0605 (0.0060) | 0.0524 (0.0065) | 0.0377 (0.0031) | 0.0351 (0.0045) | 0.0399 (0.0034) | 0.0360 (0.0056) |

Indices calculated with the offset had higher accuracies in all comparisons made (Table A8). These models were run with the simple GAM. Given the tendency for indices calculated with the offset to perform better, we carried indices with the offset forward in model development.

**Table A8.** Test MSE (and standard deviation of test MSE in parentheses) for candidate models with and without offset.

| | Sentinel-2 IA dNBR | | Sentinel-2 EA RBR | |
| --- | --- | --- | --- | --- |
| | **With Offset** | **No Offset** | **With Offset** | **No Offset** |
| CBI | 0.0185 (0.0020) | 0.0193 (0.0018) | 0.0240 (0.0018) | 0.0253 (0.0019) |
| ΔBA | 0.0416 (0.0069) | 0.0428 (0.0065) | 0.0416 (0.0059) | 0.0443 (0.0064) |
| Adj. FVS ΔCC | 0.0351 (0.0045) | 0.03670 (0.0041) | 0.0340 (0.0039) | 0.0368 (0.0042) |

Note that, in our study, we developed separate equations for EA versus IA timeframes, rather than applying a correction factor to EA models to arrive at IA predictions [45]. Limitations with this approach and other post-fire fire effects studies should be considered when using these models. Our approach of developing models for both EA and IA timelines using only EA data carries the assumption that burn severity and stand metrics are roughly similar immediately post-fire versus one growing season post-fire. For the strata which are most likely to change from IA to EA post-fire timelines, the CBI methodology includes survey questions that would moderate shifts in CBI such as the presence of colonizers and change in species composition in the understory strata as well as char height for the trees, which should stay the same immediately versus 1-year post-fire. Most of the other metrics apply to fire effects as they occur due to the fire, not how they abate after 1 year. Basal area should remain similar between EA and IA and may be easier to determine 1-year post-fire because fire-killed trees would likely not have foliage.

## Appendix B

*Appendix B.1. Final Models*

Appendix B.1.1. Final Model Coefficients and Equations

Model: CBI for IA timeframe
Predictor variable: Sentinel dNBR with offset
Model statement: gamlss(formula = CBI ~ dNBR, sigma.formula $\cong$ dNBR, nu.formula $\cong$ dNBR, tau.formula $\cong$ dNBR, family = BEINF, data = na.omit(SWRAVG_field))
Coefficients (intercept, predictor):
Mu: −1.033641, 0.005051
Sigma: −0.47943, −0.00123
Nu: 1.09289, −0.04033
Tau: −9.479199, 0.008912

Model: ΔBA for IA timeframe
Predictor variable: Sentinel dNBR with offset
Model statement: gamlss(formula = ΔBA ~ dNBR, sigma.formula $\cong$ dNBR, nu.formula $\cong$ dNBR, tau.formula $\cong$ dNBR, family = BEINF, data = na.omit(SWRAVG_field))
Coefficients (intercept, predictor):
Mu: −2.329664 0.005388
Sigma: −0.238895 0.001175
Nu: 1.71349 −0.01886
Tau: −4.591958 0.009354

Model: ΔCC for IA timeframe
Predictor variable: Sentinel dNBR with offset
Model statement: gamlss(formula = ΔCC ~ dNBR, sigma.formula $\cong$ dNBR, nu.formula $\cong$ dNBR, tau.formula $\cong$ dNBR, family = BEINF, data = na.omit(SWRAVG_field))
Coefficients (intercept, predictor):
Mu: −1.834267 0.005703
Sigma: −0.5793095 −0.0008575
Nu: 1.27214 −0.02225
Tau: −5.17080 0.01224

Model: CBI for EA timeframe
Predictor variable: Sentinel RBR with offset
Model statement: gamlss(formula = CBI ~ RBR, sigma.formula $\cong$ RBR, nu.formula $\cong$ RBR, tau.formula $\cong$ RBR, family = BEINF, data = na.omit(SWRAVG_field))
Coefficients (intercept, predictor):
Mu: −0.995575 0.008016
Sigma: −0.52598 −0.00168
Nu: 0.22578 −0.04363
Tau: −18.91817 0.03696

Model: ΔBA for EA timeframe
Predictor variable: Sentinel RBR with offset
Model statement: gamlss(formula = ΔBA ~ RBR, sigma.formula $\cong$ RBR, nu.formula $\cong$ RBR, tau.formula $\cong$ RBR, family = BEINF, data = na.omit(SWRAVG_field))
Coefficients (intercept, predictor):
Mu: −2.387856 0.008696
Sigma: −0.359833 0.002062
Nu: 1.28024 −0.02816
Tau: −4.62454 0.01483

Model: ΔCC for EA timeframe

Predictor variable: Sentinel RBR with offset
Model statement: gamlss(formula = ΔCC ~ RBR, sigma.formula ≅ RBR, nu.formula ≅ RBR, tau.formula ≅ RBR, family = BEINF, data = na.omit(SWRAVG_field))
Coefficients (intercept, predictor):
Mu: −1.773280 0.008446
Sigma: −0.714907 0.001485
Nu: 0.8161 −0.0338
Tau: −4.71010 0.01688

Appendix B.1.2. Final Model Confusion Matrices

Confusion matrices for each of the final models are presented in Tables A9–A20 below.

**Table A9.** Confusion matrix for Southwest model predicting IA CBI with Sentinel-2 dNBR with offset.

| Prediction | Reference | | | | Total | User's Accuracy (%) |
|---|---|---|---|---|---|---|
| | 0–<0.1 | 0.1–<1.25 | 1.25–<2.25 | 2.25–3 | | |
| 0–<0.1 | 9 | 3 | 0 | 0 | 12 | 75.0 |
| 0.1–<1.25 | 52 | 73 | 16 | 1 | 142 | 51.4 |
| 1.25–<2.25 | 1 | 29 | 67 | 24 | 121 | 55.4 |
| 2.25–3 | 0 | 0 | 4 | 58 | 62 | 93.5 |
| Total | 62 | 105 | 87 | 83 | 337 | |
| Producer's accuracy (%) | 14.5 | 69.5 | 77.0 | 69.9 | | 61.4 |

**Table A10.** Confusion matrix for current RAVG (Miller et al., 2009) model predicting IA CBI with Landsat RdNBR with offset.

| Prediction | Reference | | | | Total | User's Accuracy (%) |
|---|---|---|---|---|---|---|
| | 0–<0.1 | 0.1–<1.25 | 1.25–<2.25 | 2.25–3 | | |
| 0–<0.1 | 61 | 87 | 39 | 1 | 188 | 32.4 |
| 0.1–<1.25 | 1 | 5 | 9 | 3 | 18 | 27.8 |
| 1.25–<2.25 | 0 | 11 | 36 | 23 | 70 | 51.4 |
| 2.25–3 | 0 | 2 | 3 | 56 | 61 | 91.8 |
| Total | 62 | 105 | 87 | 83 | 337 | |
| Producer's accuracy (%) | 98.4 | 4.8 | 41.4 | 67.5 | | 46.9 |

**Table A11.** Confusion matrix for Southwest model predicting IA BA change with Sentinel-2 dNBR with offset.

| Prediction | Reference | | | | | | Total | User's Accuracy (%) |
|---|---|---|---|---|---|---|---|---|
| | 0–<10% | 10–<25% | 25–<50% | 50–<75% | 75–<90% | 90–<100% | | |
| 0–<10% | 118 | 13 | 2 | 1 | 0 | 0 | 134 | 88.1 |
| 10–<25% | 50 | 9 | 3 | 2 | 0 | 4 | 68 | 13.2 |
| 25–<50% | 17 | 9 | 12 | 11 | 5 | 3 | 57 | 21.1 |
| 50–<75% | 0 | 6 | 3 | 2 | 1 | 14 | 26 | 7.7 |
| 75–<90% | 0 | 0 | 0 | 2 | 2 | 12 | 16 | 12.5 |
| 90–<100% | 0 | 0 | 0 | 1 | 2 | 33 | 36 | 91.7 |
| Total | 185 | 37 | 20 | 19 | 10 | 66 | 337 | |
| Producer's accuracy (%) | 63.8 | 24.3 | 60.0 | 10.5 | 20.0 | 50.0 | | 52.2 |

**Table A12.** Confusion matrix for current (Miller et al., 2009) model predicting IA BA change with Landsat RdNBR with offset.

| | Reference | | | | | | | |
|---|---|---|---|---|---|---|---|---|
| Prediction | 0–<10% | 10–<25% | 25–<50% | 50–<75% | 75–<90% | 90–<100% | Total | User's Accuracy (%) |
| 0–<10% | 167 | 21 | 9 | 2 | 0 | 2 | 201 | 83.1 |
| 10–<25% | 10 | 2 | 4 | 1 | 1 | 3 | 21 | 9.5 |
| 25–<50% | 3 | 6 | 3 | 6 | 3 | 3 | 24 | 12.5 |
| 50–<75% | 2 | 3 | 1 | 5 | 3 | 4 | 18 | 27.8 |
| 75–<90% | 0 | 5 | 2 | 1 | 0 | 4 | 12 | 0.0 |
| 90–<100% | 3 | 0 | 1 | 4 | 3 | 50 | 61 | 82.0 |
| Total | 185 | 37 | 20 | 19 | 10 | 66 | 337 | |
| Producer's accuracy (%) | 90.3 | 5.4 | 15.0 | 26.3 | 0.0 | 75.8 | | 67.4 |

**Table A13.** Confusion matrix for Southwest model predicting IA scorch-adjusted canopy cover change with Sentinel-2 dNBR with offset.

| | Reference | | | | | | | |
|---|---|---|---|---|---|---|---|---|
| Prediction | 0–<10% | 10–<25% | 25–<50% | 50–<75% | 75–<90% | 90–<100% | Total | User's Accuracy (%) |
| 0–<10% | 69 | 13 | 4 | 0 | 0 | 0 | 86 | 80.2 |
| 10–<25% | 26 | 23 | 16 | 3 | 0 | 0 | 68 | 33.8 |
| 25–<50% | 8 | 27 | 31 | 7 | 4 | 8 | 85 | 36.5 |
| 50–<75% | 0 | 4 | 10 | 2 | 3 | 8 | 27 | 7.4 |
| 75–<90% | 0 | 1 | 4 | 3 | 1 | 12 | 21 | 4.8 |
| 90–<100% | 0 | 0 | 0 | 1 | 4 | 45 | 50 | 90.0 |
| Total | 103 | 68 | 65 | 16 | 12 | 73 | 337 | |
| Producer's accuracy (%) | 67.0 | 33.8 | 47.7 | 12.5 | 8.3 | 61.6 | | 50.7 |

**Table A14.** Confusion matrix for current (Miller et al., 2009) model predicting IA canopy cover change with Landsat RdNBR with offset.

| | Reference | | | | | | | |
|---|---|---|---|---|---|---|---|---|
| Prediction | 0–<10% | 10–<25% | 25–<50% | 50–<75% | 75–<90% | 90–<100% | Total | User's Accuracy (%) |
| 0–<10% | 98 | 40 | 15 | 1 | 0 | 0 | 154 | 63.6 |
| 10–<25% | 1 | 8 | 8 | 3 | 0 | 1 | 21 | 38.1 |
| 25–<50% | 2 | 10 | 15 | 2 | 0 | 2 | 31 | 48.4 |
| 50–<75% | 2 | 4 | 13 | 3 | 2 | 8 | 32 | 9.4 |
| 75–<90% | 0 | 1 | 5 | 5 | 3 | 4 | 18 | 16.7 |
| 90–<100% | 0 | 5 | 9 | 2 | 7 | 58 | 81 | 71.6 |
| Total | 103 | 68 | 65 | 16 | 12 | 73 | 337 | |
| Producer's accuracy (%) | 95.1 | 11.8 | 23.1 | 18.8 | 25.0 | 79.5 | | 54.9 |

**Table A15.** Confusion matrix for Southwest model predicting EA CBI with Sentinel-2 RBR with offset.

| | Reference | | | | | |
|---|---|---|---|---|---|---|
| Prediction | 0–<0.1 | 0.1–<1.25 | 1.25–<2.25 | 2.25–3 | Total | User's Accuracy (%) |
| 0–<0.1 | 1 | 0 | 0 | 0 | 1 | 100 |
| 0.1–<1.25 | 61 | 87 | 26 | 1 | 175 | 49.7 |
| 1.25–<2.25 | 0 | 18 | 56 | 17 | 91 | 61.5 |
| 2.25–3 | 0 | 0 | 5 | 65 | 70 | 92.9 |
| Total | 62 | 105 | 87 | 83 | 337 | |
| Producer's accuracy (%) | 1.6 | 82.9 | 64.4 | 78.3 | | 62.0 |

**Table A16.** Confusion matrix for current (Miller et al., 2009) model prediction EA CBI with Landsat RdNBR with offset.

| | | | Reference | | | |
|---|---|---|---|---|---|---|
| **Prediction** | **0–<0.1** | **0.1–<1.25** | **1.25–<2.25** | **2.25–3** | **Total** | **User's Accuracy (%)** |
| 0–<0.1 | 60 | 83 | 28 | 1 | 172 | 34.9 |
| 0.1–<1.25 | 2 | 8 | 15 | 2 | 27 | 29.6 |
| 1.25–<2.25 | 0 | 10 | 37 | 16 | 63 | 58.7 |
| 2.25–3 | 0 | 4 | 7 | 64 | 75 | 85.3 |
| Total | 62 | 105 | 87 | 83 | 337 | |
| Producer's accuracy (%) | 96.8 | 7.6 | 42.5 | 77.1 | | 50.1 |

**Table A17.** Confusion matrix for Southwest model predicting EA BA change with Sentinel-2 RBR with offset.

| | | | | Reference | | | | |
|---|---|---|---|---|---|---|---|---|
| **Prediction** | **0–<10%** | **10–<25%** | **25–<50%** | **50–<75%** | **75–<90%** | **90–<100%** | **Total** | **User's Accuracy (%)** |
| 0–<10% | 131 | 13 | 1 | 0 | 0 | 0 | 145 | 90.3 |
| 10–<25% | 39 | 7 | 5 | 2 | 0 | 4 | 57 | 12.3 |
| 25–<50% | 12 | 11 | 10 | 6 | 5 | 4 | 48 | 20.8 |
| 50–<75% | 3 | 3 | 3 | 8 | 0 | 14 | 31 | 25.8 |
| 75–<90% | 0 | 3 | 1 | 1 | 3 | 14 | 22 | 13.6 |
| 90–<100% | 0 | 0 | 0 | 2 | 2 | 30 | 34 | 88.2 |
| Total | 185 | 37 | 20 | 19 | 10 | 66 | 337 | |
| Producer's accuracy (%) | 70.8 | 18.9 | 50.0 | 42.1 | 30.0 | 45.5 | | 56.1 |

**Table A18.** Confusion matrix for current (Miller et al., 2009) model predicting EA BA change with Landsat RdNBR with offset.

| | | | | Reference | | | | |
|---|---|---|---|---|---|---|---|---|
| **Prediction** | **0–<10%** | **10–<25%** | **25–<50%** | **50–<75%** | **75–<90%** | **90–<100%** | **Total** | **User's Accuracy (%)** |
| 0–<10% | 142 | 10 | 3 | 0 | 0 | 0 | 155 | 91.6 |
| 10–<25% | 9 | 8 | 2 | 2 | 0 | 1 | 22 | 36.4 |
| 25–<50% | 20 | 3 | 5 | 0 | 1 | 1 | 30 | 16.7 |
| 50–<75% | 7 | 7 | 5 | 4 | 3 | 5 | 31 | 12.9 |
| 75–<90% | 3 | 2 | 2 | 6 | 1 | 4 | 18 | 5.6 |
| 90–<100% | 4 | 7 | 3 | 7 | 5 | 55 | 81 | 67.9 |
| Total | 185 | 37 | 20 | 19 | 10 | 66 | 337 | |
| Producer's accuracy (%) | 76.8 | 21.6 | 25.0 | 21.1 | 10.0 | 83.3 | | 63.8 |

**Table A19.** Confusion matrix for Southwest model predicting EA scorch-adjusted canopy cover change with Sentinel-2 RBR with offset.

| | | | | Reference | | | | |
|---|---|---|---|---|---|---|---|---|
| **Prediction** | **0–<10%** | **10–<25%** | **25–<50%** | **50–<75%** | **75–<90%** | **90–<100%** | **Total** | **User's Accuracy (%)** |
| 0–<10% | 80 | 17 | 4 | 0 | 0 | 0 | 100 | 80.0 |
| 10–<25% | 21 | 29 | 23 | 1 | 0 | 1 | 74 | 39.2 |
| 25–<50% | 2 | 17 | 24 | 6 | 2 | 5 | 56 | 42.9 |
| 50–<75% | 0 | 5 | 11 | 3 | 4 | 11 | 34 | 8.8 |
| 75–<90% | 0 | 0 | 3 | 5 | 1 | 13 | 22 | 4.5 |
| 90–<100% | 0 | 0 | 0 | 1 | 5 | 43 | 51 | 84.3 |
| Total | 103 | 68 | 65 | 16 | 12 | 73 | 337 | |
| Producer's accuracy (%) | 77.7 | 42.6 | 36.9 | 18.8 | 8.3 | 58.9 | | 53.4 |

**Table A20.** Confusion matrix for current (Miller et al., 2009) model predicting EA canopy cover change with Landsat RdNBR with offset.

| Prediction | Reference | | | | | | Total | User's Accuracy (%) |
|---|---|---|---|---|---|---|---|---|
| | 0–<10% | 10–<25% | 25–<50% | 50–<75% | 75–<90% | 90–<100% | | |
| 0–<10% | 98 | 40 | 15 | 1 | 0 | 0 | 154 | 63.6 |
| 10–<25% | 1 | 8 | 8 | 3 | 0 | 1 | 21 | 38.1 |
| 25–<50% | 2 | 10 | 15 | 2 | 0 | 2 | 31 | 48.4 |
| 50–<75% | 2 | 4 | 13 | 3 | 2 | 8 | 32 | 9.4 |
| 75–<90% | 0 | 1 | 5 | 5 | 3 | 4 | 18 | 16.7 |
| 90–<100% | 0 | 5 | 9 | 2 | 7 | 58 | 81 | 71.6 |
| Total | 103 | 68 | 65 | 16 | 12 | 73 | 337 | |
| Producer's accuracy (%) | 95.1 | 11.8 | 23.1 | 18.8 | 25.0 | 79.5 | | 54.9 |

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
