# Peer review of "Region-Specific Remote-Sensing Models for Predicting Burn Severity, Basal Area Change, and Canopy Cover Change following Fire in the Southwestern United States"

_fire, doi:10.3390/fire5050137_

Round 1

Reviewer 1 Report

General

Let me apologize for providing a somewhat incomplete report.
One reason is that my impression is that the manuscript needs quite a significant revision before the acceptance.
I would be glad to carry out a more complete review for a possible next version of the manuscript.

The authors have designed and conducted  a comprehensive study and carried out thorough work with several methods.
Some advanced methods have been used but the manuscript looks like a study report. One of the final outcomes, "region-specific post-wildfire model had several advantages over conventional California-based models", should be quite obvious without any study.
It is a bit hard to find novelties and new ideas in the methods and approaches of the study.

Mapping of fire risks and fire severity are important,
but in addition to map form products, areal estimates and their error estimates
would also be interesting, such as the the estimates of the risk areas and the burnt area and areas  of canopy cover and basal area losses in different categories after the fire.
Statistically sound methods to estimate those areas with the error estimates
would make the article scientifically more interesting.

The general conclusions and their reliability are affected also many types
of uncertainties, e.g., sampling strategies, measurement errors and modeling,
as the authors themselves also discuss.
A overall error budget would be very interesting and support the conclusions
but is also challenging.

Finally, the authors discuss the directions for future research in the subsection 4.3  in a nice way, e.g., those regarding the use of concomitant field data and
"contemporary statistical learning tools as well as meta-analyses using combined data, or the use of similar datasets as validation sets for model development".
A reader may wonder why some of them have not been done already in the current study although the lack of resources could be an excuse.
Could you further comment this?

Some further general comments
1
A list of the study phases and the date sets in a form of a flowchart or just as a list would help reading.
It could be, e.g.,  in the beginning of the section 2 Methods.

2
How the different sampling schemes and estimation phases affect the error estimates would be an interesting question, see above.

3
I am wondering would it help in reading the article if the authors could write the equations of the Satellite image based indices, e.g., in Section 2.3, although a reference is given.

4

The effect of inaccurate location of the field plots on the estimates can be reduced using smoothing as presented in Section 2.3. It needs subjective weights for the neighboring pixels. Could you please comment the selection of the weights.

5 L 162-170
"Circular plots measured 30 m in diameter and were located at least 500 m apart, ≥100 m from roads or trails, in areas with >10% tree cover, and in areas of homogeneous burn severity, preferably 60m × 60m (Key and Benson 2006). ....
We included unburned plots (n=67) as 20% of
the entire dataset to ensure that models span the full range of wildfire severities (Parks et al. 2019)."
You have thus selected homogeneous field plots intentionally.
The effect on the models and conclusions has been discussed to some extent in section Discussions.
Could it be possible to argue this selection?

6
As regards photo interpretation plots, you write
"Burned photo plots were selected if they were at least
60 m inside the fire perimeter to avoid plots being partially in or out of the
fire. Unburned photo plots were identified within the fire perimeter and also
within a 500 m buffer outside of the fire perimeter to allow for approximately
half of unburned plot sampling to occur outside the fire perimeter."
On the other hand, in the section Discussion, you write
" ...whereas PI plots
were located on a systematic grid which could have increased the ratio of PI
plots in areas of mixed severity..." (L499-500).
Could you please clarify?
A flowchart or a list of the study phases could help the understanding.

7
Please list all variables and abbreviations in Table 4 (or earlier in the text).
E.g., CBI is not defined there, only CBI.B (derived from CBI).

A few detailed comments
L 26-27
"from southwest U.S. data had clear advantages compared to the current CA-based models"
CA-based is not defined earlier.

L 43-46
"Fire frequency typically increases in
the Southwest with elevation and moisture, and fires often become stand-replacing  at higher elevations, occurring mainly during extreme drought
(Hurteau et al. 2014)."
Clarify this please, or, may be the problem is my poor understanding of the conditions.

L71-88
Could you present these by equations although your are citing the previous studies, see also later.

L 161-162
"Sampling was done at even intervals along roads or trails at a target density of 15 - 30 plots per fire."
A question: Does the closeness of a road or trail affect the randomness
(objectiveness or representativeness) of the sample plots?

L 259-260
"to retain up to 40 PI plots per fire after stratification and
exclusion of plots due to low canopy cover or edge effects."
Please tell the details of stratification.

L 349-355
A good practice is to list (define) all notations after the equations.

L 468
"with potential co-linearity between variables"
Do you mean collinearity?

Reviewer 2 Report

The article deals with the burn severity, basal area change and canopy cover change following fires in the southwestern United States. It is an interesting approach, well-written and easy to follow.

In the introduction some more comments are needed to better state the wildfire problem in a wider context with well-documented references:

in highly fire-prone ecosystems, loss of biodiversity, ecosystem function or services after wildfire events occurring with unnaturally high frequency, the magnitude of extent or intensity can result in land degradation or even complete transformation of the ecosystem. Besides their impacts on the carbon cycle, such events, usually called megafires because of their size, reduce the amount of living biomass, affect species composition, water and nutrient cycles, increase flood risk and soil erosion, and threaten local livelihoods by burning agricultural land and homes. In addition, these fires have devastating impacts on local wildlife as animals either are unable to escape from the fires or become threatened by the loss of their habitat, food, and shelter. Also, in the aftermath of wildfire accelerated erosion (Stefanidis et al. 2022) and flash flood phenomena occur (Wilder et al. 2021).

Stefanidis, S., Alexandridis, V., Spalevic, V., & Mincato, R. L. (2022). Wildfire Effects on Soil Erosion Dynamics: The Case of 2021 Megafires in Greece. Agriculture & Forestry, 68(2), 49-63.

Wilder, B. A., Lancaster, J. T., Cafferata, P. H., Coe, D. B., Swanson, B. J., Lindsay, D. N., ... & Kinoshita, A. M. (2021). An analytical solution for rapidly predicting post‐fire peak streamflow for small watersheds in southern California. Hydrological Processes, 35(1), e13976.

Line 117. How the dNBR or relative products can be used in post-fire management? It would be interesting to have some more information.

Line 146. How that vegetation types fit with fuel types models?

The targets for future research are very comprehensive and highlight the importance of this research. Also, it could be added a statement as the ever-growing availability of earth observation (EO) data and the well-established use of geospatial technologies could help the creation of automated workflows for wildfire management and predictions.

It is preferable to have the conclusion as a separate chapter rather than included as a sub-chapter of the discussion.

In general, this is a very interesting article and highly important for the readers of the journal. Congratulation to the authors for the article. My suggestions in the aforementioned parts I believe will strengthen the scientifically sound/representation of the research.

Reviewer 3 Report

Thank you for the invitation. The authors implemented the use of several indices for remotely sensed images in order to define the burn severity and temporal changes of basal area and canpoy cover between pre-fire and post-fire periods. The results look encouraging and motivating, which makes this manuscript valuable. Data used, ground-truthing, used indices, and performance of the tasks are clearly defined. The discussions and conclusions are sufficient. I didn't detect any language or ethical issues. I recommend a minor revision before printing. I have listed my recommendations below.
-There are some typos in the text. Please control them.

-Please add (CA), (AZ), and (NM) after the first use of California, Arizona, and New Mexico in the abstract and introduction. Add the long forms of PI, KNN, and MSE before using them abbreviated for the first time.

-The vegetation information for the study area described between Lines 135-145 requires relevant citations.

-Please use SI units throughout the article. I see acres in the intro and miles on map scales.

-Please mention what sort of L8 OLI images you used. Surface reflectance or top-of-atmosphere reflectance? We need to understand the atmospheric correction phase.

- I’m not satisfied with the table representation of filter kernels (Tables 2 and 3). Please try to represent them as matrices.

-Please check Line 247. There is a referencing error.

-Line 303: You use several topographic factors. Therefore, you need to give information about the DEM/DTM you used.

-Please explain in the text what a, b, and index denote in Eqs. 1-2-3.

-The use of active fire pixels (e.g., FIRMS) can be added as a future study in the conclusions.

Round 2

Reviewer 1 Report

The authors have revised the manuscript based on my general and detailed
comments or given acceptable rebuttals.

I do not have any further comments.

I am glad to recommend the acceptance of the manuscript.

Reviewer 2 Report

The article is now accepted for publication